# Sex Hormones and Their Effects on Ocular Disorders and Pathophysiology: Current Aspects and Our Experience

**DOI:** 10.3390/ijms23063269

**Published:** 2022-03-17

**Authors:** Raffaele Nuzzi, Paolo Caselgrandi

**Affiliations:** Eye Clinic Section, Department of Surgical Sciences, University of Turin, 10126 Turin, Italy; paolo.caselgrandi@unito.it

**Keywords:** cornea, dry eye, estrogens, eye, glaucoma, ocular disorders, optic nerve, sex hormones

## Abstract

Sex hormones are molecules produced by the gonads and to a small extent by the adrenal gland, which not only determine the primary and secondary sexual characteristics of an individual, differentiating man from woman, but also participate in the functioning of the various systems of the body. The evidence that many eye diseases differ in terms of prevalence between men and women has allowed us, in recent years, to carry out several studies that have investigated the association between sex hormones and the pathophysiology of eye tissues. Specific receptors for sex hormones have been found on the lacrimal and meibomian glands, conjunctiva, cornea, lens, retina, and choroid. This work summarizes the current knowledge on the role that sex hormones play in the pathogenesis of the most common ocular disorders and indicates our clinical experience in these situations. The aim is to stimulate an interdisciplinary approach between endocrinology, neurology, molecular biology, and ophthalmology to improve the management of these diseases and to lay the foundations for new therapeutic strategies.

## 1. Introduction

There are differences between men and women in many eye disorders. This observation has allowed, in recent years, the development of studies that have investigated the type of relationship between sex hormones and various eye diseases. It is in fact widely known that sex hormones play a key role not only at the level of the reproductive organs, where the differences between sexes are evident, but also at the level of many other organs and tissues that are apparently identical between men and women. The steroid sex hormones, being conveyed by the bloodstream, reach all the tissues of the body; however, they only exert their effect on cells that are equipped with specific receptors. An important role of sex hormones, for example, has been recognized at the level of the central nervous system, the cardiovascular system, and the osteo-articular system [1,2,3]. At the eye level, these hormones were found to influence the regulation of the blood–retinal barrier, neuroprotection [4,5], vascular regulation [6,7], and the modification of intraocular pressure values [8,9], leaving room for hypotheses of their role in the pathogenesis of many ocular disorders. Specific receptors for sex hormones have been identified in the lacrimal glands, meibomian glands, bulbar and eyelid conjunctiva, cornea, iris and ciliary body, lens, and at the level of the retina and choroid [10]. In two of our previous works, we highlighted the relationships existing between sex hormones and respectively retinal pathologies [11] and pathologies affecting the optic nerve [12]. This paper aims to indicate the role that sex steroid hormones play in the pathogenesis of the most common ocular disorders as well as the possible interdisciplinary clinical-therapeutic perspectives.

## 2. Sex Hormones

### 2.1. Steroidogenesis

The main classes of sex steroids are androgens, estrogens, and progestogens. They are produced in the testicles and ovaries and in a small part also by the adrenal gland starting from cholesterol, which is the primary precursor of steroidogenesis. The entire process is controlled by gonadotropins, hormones produced in the adenohypophysis, which are in turn regulated by pituitary control [13]. Cholesterol can be taken from the bloodstream, from cell deposits, or synthesized in situ by cells. Steroidogenesis begins in the mitochondria where most of the necessary enzymes are present such as cytochrome P450, 3β-hydroxysteroid dehydrogenase, and ferredoxin reductase [14,15]. Cholesterol, after being converted to pregnenolone, is transformed into dehydroepiandrosterone (DHEA) and androstenedione. DHEA is the common precursor for the biosynthesis of both testosterone, the main androgen circulating in humans, and of the three primary estrogens, 17β-estradiol, estrone, and estriol. Estrogens derive from androgens thanks to the action of the aromatase enzyme. Estrogens and androgens are mainly produced by the gonads and cortex of the adrenal gland, but are also produced by other tissues that possess specific steroidogenic enzymes such as adipose tissue or the liver, starting with common substrates such as DHEA and androstenedione. This peripheral production of sex hormones affects both sexes, but is of fundamental importance for women. In fact, while for men, the production of androgens by the testicles remains high over the years, for women, the amount of estrogen produced by the ovaries stops with menopause and the only source of sex hormones remains extragonadal production [16]. Progesterone is produced by the corpus luteum and the placenta and is important for regulating the menstrual cycle, pregnancy, and breastfeeding.

### 2.2. Mechanism of Action

Steroid sex hormones can exert their action on target cells through two different mechanisms, a classical genomic pathway and a non-genomic pathway [17].

The classical genomic route is slower and lasts from hours to days. It foresees that from the bloodstream, the sex hormones diffuse thanks to a passive transport through the plasma membrane and reach the cellular cytoplasm where they bind, forming complexes with the specific intracellular receptors, activating them. The activated ligand–receptor complex, therefore, acts as a transcription factor by binding to a specific DNA sequence in the cell nucleus, first allowing the transcription of specific mRNAs, and subsequently, the production of the related proteins.

The non-genomic route is faster and lasts from seconds to minutes. It was discovered many years later than the genomic pathway and allowed us to explain the effects induced by sex hormones that previously could not be explained [18]. This pathway requires the sex hormones to bind to specific membrane proteins bound to G-proteins, which lead to the activation of the membrane protein phospholipase C (PLC), which in turn leads to the formation of inositol trisphospate (IP3). IP3 binds to specific receptors at the level of the smooth endoplasmic reticulum, resulting in the release of calcium and the activation of further signaling pathways. The non-genomic route is therefore faster and involves the opening or closing of ion channels and altering the functioning of transport proteins [19,20].

Estrogen receptors are encoded by genes located on chromosome 14 [21], while androgen receptors are encoded by genes on chromosome X [22].

Sex hormones can exert their effect in an endocrine, paracrine, or autocrine way, depending on whether they act respectively on a distant cell belonging to another tissue or organ, on a neighboring cell of the same tissue or organ, or on the same cell that has products. The paracrine or autocrine signaling pathway is particularly important in postmenopausal women, where sex hormones have often been shown to act on the same peripheral tissues from which they were produced [16,23].

Through these mechanisms of action, sex hormones not only determine the primary and secondary sexual characteristics of an individual, but also participate in the functioning of the various systems of the body. For example, estrogens are essential for the regulation of metabolism, bone mineralization, and vasoprotection [24]; androgens, on the other hand, control mood, desire and sexual instincts, muscle and bone development, and erythropoiesis [25].

## 3. Effects of Sex Hormones on Ocular Disorders

Figure 1 and Table 1 list the ocular disorders for which there is evidence of a role of sex hormones in their pathophysiology. In clinical practice, the involvement of sex hormones in these disorders presents an interpersonal variability and must always be evaluated in an interdisciplinary endocrine–ophthalmological context.

There are other eye diseases that could be affected by sex hormones such as thyroid associated ophthalmopathy (TAO), idiopathic choroidal neovascularization, or uveitis associated with arthritis, but to date, there are no studies and scientific evidence in this regard.

### 3.1. Dry Eye Disease

Dry eye disease (DED) is a multifactorial disease affecting the tear film and ocular surface that involves symptoms of ocular discomfort, visual disturbances, and lubrication problems that can also lead to damage to the cornea. DED is accompanied by an increased osmolarity of the tear film and inflammation of the ocular surface [26]. DED can be classified into two categories based on the etiopathogenetic mechanism: hyposecretive, characterized by the decreased production of tears by the lacrimal glands, and evaporative, characterized by the qualitative alteration and instability of the tear film with a greater tendency to evaporation due to an altered activity of the meibomian glands [27].

The main symptoms reported by patients are foreign body sensation, burning, photophobia, blurred vision, and itching. Diagnosis is achieved through the quantification of symptoms with the use of validated questionnaires such as the Ocular Surface Disease Index (OSDI), the assessment of tear film instability with the measurement of break up time (BUT) with fluorescein, measurement of tear osmolarity, measurement of tear volume with Schirmer’s test, and ocular and conjunctival surface staining with fluorescein and lissamine green, respectively.

The prevalence of DED shows that women are affected by two to four times more than men of the same age [26,28] and this difference is accentuated after menopause [29]. These data suggest that sex hormones play a role in the pathogenesis of the disease [30,31,32]. The estrogen and androgen receptors were found to be present at the level of all the tissues that make up the ocular surface: on the conjunctiva, cornea, and at the level of the lacrimal and meibomian glands [33]. Various studies have shown that steroid sex hormones contribute to maintaining the balance and function of the ocular surface, although the precise mechanisms are not yet clear [25].

Estrogens and androgens appear to have opposite effects in the pathogenesis of DED.

The role of estrogen is not yet fully understood, but several in vitro animal and human studies show that estrogens decrease the secretion of the sebaceous glands, inhibit lipogenesis, and promote inflammation of the ocular surface. The basic mechanism could be the competitive link that estrogens exert on androgen receptors, thus inhibiting the activity of the meibomian glands induced by androgens [34]. Another explanation could be that estrogen downregulates the cyclic AMP signaling pathway, which is responsible for cell proliferation in the meibomian glands [35]. Clinically, the data seem conflicting. Many studies describe a worsening of DED symptoms with high estrogen levels, as occurs during the estrogenic peak in the follicular phase of the menstrual cycle, thus highlighting a pro-inflammatory role of estrogen. Many other evidence, on the other hand, point to a progressive increase in symptoms with low estrogen levels, as occurs during menopause. Menopause represents the end of a woman’s reproductive life and is determined after 12 months of amenorrhea [36]. In this phase, there is a progressive decrease in the blood levels of estrogen and progestogen, but also of androgens. Many works in the literature indicate that menopause is associated with an increased risk of DED, mainly of the evaporative type with dysfunction of the meibomian glands [37,38]. However, some studies have shown that the use of systemic hormonal replacement treatments (SHRT) in menopausal women increases the risk of DED [39].

Although the role of estrogen in relation to the ocular surface appears controversial, the hypothesis has recently been advanced that during menopause, evaporative DED is not so much due to the reduction in estrogen, but rather to the reduction in androgens [40,41].

Unlike estrogens, androgens promote the activity of the meibomian glands, stimulating the lipogenesis and maturation of acinar cells [41,42], and have an anti-inflammatory role, favoring the synthesis of TGF-β and inhibiting that of interleukin-1β and TNF-α [43]. On a clinical level, in fact, there is an alteration in the functioning of the meibomian glands and an increase in the risk of DED in all cases in which there is a reduction in circulating androgens, a dysfunction or insensitivity of the androgen receptors, or anti-androgen therapy [44,45]. This evidence was confirmed in our clinical experience, in which patients treated for prostate cancer with anti-androgen drugs who come complaining of burning, photophobia, and foreign body sensation are often affected by DED, especially of the evaporative type.

Although the meibomian glands possess receptors for both estrogen and androgen, it is the latter that are of fundamental importance in determining the quantity and quality of tear production [46].

### 3.2. Corneal Disorders

The cornea is an avascular and transparent tissue and represents the most anterior portion of the eye. Its structure can be divided into five layers, three cellular and two acellular. The three cell layers, starting from the outermost zone, are represented by a non-keratinized stratified epithelium, a stromal matrix consisting of lamellar collagen fibers in which keratocytes are dispersed, and finally by a layer of endothelial cells. The two acellular layers are instead the Bowman’s membrane, interposed between the epithelium and the stroma, and the Descemet’s membrane, interposed between the stroma and the endothelium [47]. If the normal cytoarchitecture of the cornea is subverted, as often happens in corneal disorders or pathologies, its refractive and biomechanical characteristics are consequently altered. The tear film that lines the ocular surface is essential for correct corneal metabolism not only because it performs a lubrication and anti-microbial protection function, but also because it allows the diffusion of oxygen, nutrients, growth factors, and hormones [48].

The observation of structural changes in the cornea during the normal menstrual cycle and during pregnancy has suggested that sex hormones may also play an important role at the corneal level. Various studies have confirmed that estrogen, androgen, and progestogen receptors are present in the epithelium, stroma, and corneal endothelium [49,50].

Changes in corneal thickness, curvature, and sensitivity have been observed during the menstrual cycle. In particular, an increase in corneal thickness was recorded during the ovulatory phase and the premenstrual luteal phase, respectively, at the peak of estrogen and progestogen [51,52]. In fact, these sex hormones can induce water retention in the cornea such as to increase the central corneal thickness and sometimes cause changes in the curvature of the cornea, increase in intraocular pressure and visual disturbances. These effects are mainly determined by estrogens, which stimulate the activation of proteinases of the stromal matrix and collagenolytic enzymes, responsible for the biomechanical and structural alteration of the cornea. Estrogens also promote the deposition of hyaluronic acid and hydration, leading to an increase in central corneal thickness [53,54]. Such biomechanical alterations are similarly found during pregnancy, especially during the third trimester.

These biomechanical changes of the cornea induced by fluctuations in sex hormones are evident in clinical practice. In our experience, they are particularly important in the surgical and para-surgical treatment of refractive defects, in which it is essential that there is a stability of the hormonal structure to obtain satisfactory, long-lasting, and stable post-operative results, without the risk of short-term reprocessing. For this reason, it is occasionally necessary to wait for the contraceptive pill or any hormone-based treatments to be stopped before proceeding with the treatment.

Recently, a possible role of sex hormones in the pathogenesis of keratoconus has also been hypothesized, a pathology in which there is a progressive wearing out of the corneal thickness, causing important visual disturbances. In fact, some studies have reported a significant exacerbation of the progression of keratoconus during pregnancy, corresponding to the increase in the level of sex hormones [55,56,57,58]. To date, however, it is not yet clear how these hormones can influence the progression of keratoconus [59].

Estrogen also appears to play a role in corneal wound healing. It has been shown in vitro that 17β-estradiol is able to promote the migration and proliferation of corneal epithelial cells and to stimulate the production of epidermal growth factor (EGF), which has been found to be an important corneal mediator in wound healing [60,61,62].

### 3.3. Cataract

Many studies have reported an increased incidence of cataracts in women compared to men of the same age. However, this difference between the sexes is only present after menopause [63,64], and appears to be caused by the decline in the level of estrogen in women after menopause. In fact, it has been shown that estrogens have a protective role against lens opacification, reducing cataractogenesis [65,66]. Estrogens have an antioxidant function, counteracting the formation of TGFβ, and reducing oxidative stress on the lens, which is one of the main pathogenic factors for the formation of cataracts [67,68,69]. They also regulate the hydration and ionic composition of the lens, helping to keep it transparent [19].

Numerous studies have indicated that postmenopausal SHRT, early menarche, and late menopause, in which there is increased exposure to estrogen, are found to be a protective factor and are associated with a lower risk of developing cataracts in advanced age [70].

### 3.4. Glaucoma and Optic Nerve Disorders

The association between sex hormones and optic nerve disorders has been thoroughly investigated in our previous work, which focused specifically on this topic [12].

In the literature, there are various studies that highlight a link between sex hormones and important pathologies of the optic nerve including glaucoma, Leber’s hereditary optic neuropathy (LHON), and optic neuritis.

Other studies have associated sex hormones with the onset of other optic nerve disorders such as non-arteritic anterior ischemic optic neuropathy (NAION), or with the formation of gliomas and meningiomas; however, the evidence present at the time does not allow for the establishment of any significant relationship.

#### 3.4.1. Glaucoma

Glaucoma is a neurodegenerative disease characterized by the progressive loss of retinal ganglion cells (RGCs), the thinning of the retinal nerve fiber layer (RNFL), and structural changes affecting the optic nerve [71]. Glaucoma causes progressive vision loss and is currently a leading cause of blindness worldwide. It is estimated that there will be around 112 million people with glaucoma by 2040 [72].

The two main subtypes of glaucoma are represented by primary angle closure glaucoma (PACG) and primary open angle glaucoma (POAG), defined by the anatomy of the iridocorneal angle present in the anterior chamber of the eye. JCAC accounts for approximately 26% of all forms of glaucoma and is prevalent in Asia, while POAG accounts for approximately 74% of glaucoma cases and is the prevalent form in the U.S., Europe, Africa, and Australia [73]. Common risk factors for both types are age, ethnicity, and familiarity; gender and the presence of refractive defects are specific risk factors for the PACG, while elevated intraocular pressure (IOP) is specific for POAG [74,75]. POAG is further divided into two forms, high-tension and low-tension glaucoma, depending on whether it is associated with an increase in IOP or not.

Several population studies have shown that women represent approximately 60% of glaucoma patients [73,76]. These data are partly explained by the fact that life expectancy is longer in women and by the fact that they are more affected by PACG than men for an anatomical factor, as on average, female eyes are shorter and have a narrower anterior chamber [9]. However, recent studies suggest that sex hormones, particularly estrogen, play a role in the pathogenesis of glaucoma and help explain the different distribution of this disease in the two sexes [77,78,79]. In particular, growing evidence indicates that a lower estrogen level is associated with an increased risk of developing glaucoma [80,81,82]. From various in vitro studies, estrogens appear to exert a retinal neuroprotective action by promoting the survival of RGCs, preserving the thinning of RNFL [83] and contributing to the lowering of IOP [84]. Furthermore, estrogens have been shown to have an anti-inflammatory action by promoting the downregulation of cytokine production, counteracting one of the possible mechanisms underlying the damage of the optic nerve [85,86].

These hypotheses are confirmed by the fact that in postmenopausal women, in which there is a decrease in the level of circulating estrogen, there is a significant increase in the incidence of glaucoma and an increase in IOP of about 1.5–3.5 mmHg compared to premenopausal women of the same age [87]. Similarly, several studies have shown that longer exposure to estrogen, due to early menarche and/or late menopause, is associated with a lower risk of developing glaucoma [81,82].

In clinical practice, the concentration of estrogens in glaucoma patients could represent an indication for the optimization of anti-glaucoma therapy. Estrogen could also represent an interesting treatment perspective for glaucoma [88], especially in the form of low-tension POAG, which currently does not include any therapy targets [89]. In an experimental animal model on mice in which a high IOP was surgically induced, treatment with estrogen demonstrated a significant neuroprotective effect and a lower impact on visual function, reducing the apoptosis of RGCs [90]. To date, however, there is not yet sufficient preclinical and clinical evidence to support the use of these estrogen-based treatments for glaucoma.

#### 3.4.2. Leber’s Hereditary Optic Neuropathy (LHON)

Another optic nerve disorder in which a possible role of sex hormones has been investigated is LHON, the most common mitochondrial disease due to point mutations in mitochondrial DNA. LHON is a disease characterized by degeneration of RGCs, resulting in optic nerve atrophy and loss of central vision. The disease begins in early adulthood and affects men more than women [91,92]. Women are thought to be less affected by the disease because they are protected from the effect of estrogen, despite possessing the mutated gene. Some studies have shown that the administration of estrogen is able to protect the mutated cells, counteracting the high levels of reactive oxygen species and the increased apoptosis present in LHON. Estrogens would exercise this role, thanks to the activation of the ERβ receptor in the mitochondria of RGCs, activating the superoxide dismutase enzyme and resuming mitochondrial biogenesis [93,94]. These observations set the stage for the development of estrogen treatments in LHON, which we are also working on in an interdisciplinary setting.

#### 3.4.3. Optic Neuritis

Demyelinating optic neuritis is an inflammation affecting the optic nerve and is characteristic of multiple sclerosis, representing the onset symptom in one out of five cases [95]. A possible relationship between sex hormones and optic neuritis has been hypothesized thanks to the observation that women suffering from multiple sclerosis had a decrease in relapses during pregnancy. Some studies have therefore shown that estrogens have an immunomodulating effect and favor remyelination thanks to pathways starting from the ERβ receptor, which act both at the level of CD11c^+^ brain immune cells and at the level of oligodendrocytes [96,97]. Other evidence has indicated that progesterone and its derivatives are not only able to reduce the loss of myelin, but also to promote the formation of new myelin [98]. Finally, it has also been shown that testosterone, by acting on the androgen receptors of nerve cells, is able to stimulate myelin repair and exert an anti-inflammatory action [99]. This last finding was also confirmed in our clinical experience, in which it was recorded that men are affected less frequently than women. These aspects could be fundamental in the experimental therapies that we are evaluating for the biological reactivation of the optic nerve, also following repeated optic neurites in patients with multiple sclerosis.

### 3.5. Retinal Disorders

The association between sex hormones and retinal disorders has been thoroughly investigated in our previous work, which focused specifically on this topic [11].

Various studies have shown that steroidogenic enzymes are present at the retinal level and that local synthesis of neuroactive steroids occurs [100]. These steroidogenic enzymes have been found in glial cells and photoreceptors in a similar quantity to that seen in other parts of the central nervous system [101]. In particular, the inner nuclear layer of the retina appears to be the main synthesis site since the greatest number of steroidogenic enzymes is present there [102].

Sex hormones were found to have a neuroprotective action at the retinal level, mainly due to the protection of photoreceptors from damage induced by glutamate, probably mediated by the membrane estrogen receptor GPR30 [103,104]. Sex hormones are hypothesized to play a role in the pathogenesis of various retinal disorders including age-related macular degeneration (AMD), central serous chorioretinopathy (CSCR), and retinitis pigmentosa.

Some studies have also correlated the level of circulating estrogen to the risk of developing idiopathic full-thickness macular holes and diabetic retinopathy; however, the data present at the moment on these pathologies are few and conflicting in order to draw conclusions.

On this topic, it is of fundamental importance to enhance the neuro-ophthalmological interdisciplinary studies with applicative approaches, bearing in mind that the retina and the optic nerve are extroversions of the central nervous system.

#### 3.5.1. Age-Related Macular Degeneration (AMD)

AMD is an eye disease that progressively leads to the deterioration of visual function. The main risk factors are age, which is the most important factor, smoking, obesity, hypercholesterolemia, hypertension, and atherosclerosis. Gender is generally not considered a risk factor, although some studies have indicated that AMD has a higher prevalence in women than in men [105,106]. The pathogenesis of AMD is multifactorial and is characterized by the presence of oxidative damage, chronic inflammation, and the accumulation of extracellular drusen between the retinal pigment epithelium (RPE) layer and Bruch’s membrane layer. Estrogens may have a protective role against AMD thanks to their antioxidant and anti-inflammatory properties [107]. Some studies have indicated that longer exposure to estrogen or SHRT decreases the risk of developing AMD in old age [108]. Other studies align in the same direction, stating that low estrogen levels increase the risk of developing AMD [107]. Despite these data, the Royal College of Ophthalmologists guidelines for the management of AMD, based on a meta-analysis, indicate that female sex is not a risk factor for AMD and that the highest prevalence in female sex is determined by their longer life expectancy [105,109,110].

#### 3.5.2. Central Serous Chorioretinopathy (CSCR)

CSCR is an acquired retinal disease that is characterized by the presence of an exudative detachment of the retina and/or RPE. The main risk factors are united by an increased serum level of glucocorticoids and are represented by psycho-social stress, Cushing’s syndrome, steroid therapy, infections, smoking, alcohol, and pregnancy [111,112]. The pathogenesis of CSCR is not yet fully understood, but it seems to be characterized by impaired circulation and increased capillary permeability of the choroidal vessels as well as dysfunctions of the RPE [113]. Given the higher prevalence of CSCR in men, various studies have also hypothesized a role of androgens in its pathogenesis [114,115]. In fact, there is some evidence that patients undergoing exogenous testosterone therapies have a greater risk of developing the disease and that interruption of therapy leads to an improvement in the condition with the resolution of symptoms and the accumulation of subretinal fluid [116]. To date, however, the presence of few studies and contradictory evidence prevents us from drawing any conclusions.

#### 3.5.3. Retinitis Pigmentosa

The term retinitis pigmentosa refers to a group of hereditary degenerative and clinically heterogeneous retinopathies, in which genetic mutations lead to the progressive death of photoreceptors. Retinitis pigmentosa is the most common cause of hereditary blindness and is characterized at the beginning of the disease by the degeneration of the rods, while in the more advanced stages by the degeneration of the cones [117]. Therefore, the symptoms are initially represented by tunnel vision and nyctalopia, followed by a significant reduction in visual acuity up to blindness. The pathogenetic mechanism underlying the disease is not yet fully understood, and currently, no treatments are available [118]. Some studies have shown that progesterone, thanks to its antioxidant and neuroprotective properties, could promote cell survival and inhibit photoreceptor apoptosis [119]. There are currently promising results regarding the experimental treatment of retinitis pigmentosa with progesterone or its derivatives, with evidence indicating delayed cell death of photoreceptors [120,121].

Table 2 summarizes the effect that sex hormones have on various eye disorders (Table 2).

## 4. Conclusions

Some of the more common eye disorders have a different prevalence between men and women. This observation has allowed us to understand how sex hormones play a key role in the physiology of most eye tissues and how their imbalance is the basis of many eye disorders. Currently, there is scientific evidence and our daily clinical experiences regarding the involvement of sex hormones, particularly estrogen, in the pathogenesis of dry eye disease, some corneal disorders, glaucoma, Leber’s hereditary optic neuropathy, demyelinating optic neuritis, and cataract. Regarding retinal pathologies and other ocular disorders, the data present are insufficient to draw conclusions.

Interdisciplinary evaluations and approaches between endocrinology, neurology, molecular biology, and ophthalmology are of fundamental importance for the correct classification of ocular disorders and their correlation with sex hormones and to lay the foundations for new therapeutic strategies.

## Figures and Tables

**Figure 1 ijms-23-03269-f001:**
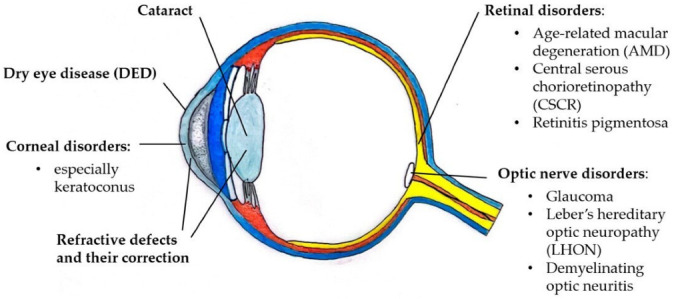
Drawing presenting the ocular disorders influenced by sex hormones. Courtesy of Dr. Sara Salomone, University of Turin, Italy.

**Table 1 ijms-23-03269-t001:** Evidence of a role of sex hormones on the pathophysiology of ocular disorders. ± = no significant evidence, + = weak evidence, ++ = strong evidence.

Ocular Disorder	Evidence of a Role of Sex Hormones on Pathophysiology	Sex Hormones Involved
Dry eye disease (DED)	++	Androgens, Estrogens
Corneal disorders	++	Estrogens
Cataract	+	Estrogens
Glaucoma	+	Estrogens
Leber’s hereditary optic neuropathy (LHON)	++	Estrogens
Demyelinating optic neuritis	++	Estrogens, Progesterone, Androgens
Age-related macular degeneration (AMD)	±	Estrogens
Central serous chorioretinopathy (CSCR)	±	Estrogens
Retinitis pigmentosa	++	Progesterone

**Table 2 ijms-23-03269-t002:** Effects of sex hormones on ocular disorders.

Ocular Disorder	Effects of Sex Hormones
Dry eye disease (DED)	Androgens promote the activity of the meibomian glands, stimulating lipogenesis and maturation of acinar cells. Androgens have an anti-inflammatory role, favoring the synthesis of TGF-β and inhibiting the synthesis of interleukin-1β and TNF-α. Estrogens inhibit cell proliferation in the meibomian glands by downregulation of the cyclic AMP signal pathway and decreasing the secretion of the sebaceous glands, inhibiting lipogenesis. Estrogens promote inflammation on the ocular surface.
Corneal disorders	Estrogens stimulate the activation of proteinases of the stromal matrix and collagenolytic enzymes, responsible for the biomechanical and structural alteration of the cornea. Estrogens promote hyaluronic acid deposition and hydration, leading to an increase in central corneal thickness. Estrogens may promote keratoconus progression (no significant current data). Estrogens promote corneal wound healing by stimulating the migration and proliferation of corneal epithelial cells and the production of epidermal growth factor (EGF).
Cataract	Estrogens have a protective role against lens opacification, reducing cataractogenesis. Estrogens have an antioxidant function, inhibiting the formation of TGFβ. Estrogens regulate the hydration and ionic composition of the lens, maintaining its transparency.
Glaucoma	Estrogens have a neuroprotective action. Estrogens promote the survival of RGCs, preserve the thinning of the RNFL, and contribute to the IOP lowering. Estrogens have an anti-inflammatory action, downregulating the cytokine production.
Leber’s hereditary optic neuropathy (LHON)	Estrogens have an antioxidant role, activating the superoxide dismutase enzyme. Estrogens reduce apoptosis of RGCs by activation of the ERβ receptor in mitochondria.
Demyelinating optic neuritis	Estrogens have an immunomodulatory effect and promote remyelination by activation of the ERβ receptor, both on CD11c^+^ brain immune cells and on oligodendrocytes. Progesterone and derivatives reduce the myelin loss and promote its regeneration. Androgens stimulate myelin repair and have an anti-inflammatory action.
Age-related macular degeneration (AMD)	Estrogens may decrease the risk of developing AMD in old age (no significant current data).
Central serous chorioretinopathy (CSCR)	Exogenous androgens may increase the risk of developing CSCR (no significant current data).
Retinitis pigmentosa	Progesterone and derivatives have an antioxidant and neuroprotective action on photoreceptors and nerve cells. Progesterone and derivatives promote cell survival and inhibit photoreceptor apoptosis.

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
