# Peer review of "Sex Hormones and Their Effects on Ocular Disorders and Pathophysiology: Current Aspects and Our Experience"

_ijms, 2022, doi:10.3390/ijms23063269_

Round 1

Reviewer 1 Report

This review is practical and has a good clinical utility. It summarizes the current knowledge on the role of sex hormones in the pathogenesis of the most common ocular disorders and and indicates the clinical experience. Though, there are a few points that the author should be noted.

  1. In table 1, the author used “+”or "++" to grade the effect on sex hormones in various ocular disorders. However,the grading factors are obscure. The author should clarify to define "+" or "++".
  2. In each section, the content is too long and the hierarchy is not clear. The author could divide the content to certain sections with subheadings such as clinical symptoms, mechanism and etc.
  3. There are some other ocular diseases which could be affected by sex hormones, such as thyroid associated ophthalmopathy (TAO), Idiopathic choroidal neovascularization, and Arthritis associated uveitis. The author should briefly mention those.
  4. A figure presenting the affected ocular diseases would make the article more much more intuitive.
  5.  

Author Response

  1. We have added a description of the meaning of the symbols ±, +, ++ in the heading of the table. They are an indicator of the strength of the evidence on the influence of sex hormones in the specific eye disorder.
  2. We accepted the suggestion and divided the longer sections (Glaucoma and optic nerve disorders, Retinal disorders) with subheadings regarding the pathologies described within them.
    In the other sections we preferred not to add subheadings because we think that the description of the disorder, the scientific evidence and our clinical experience are a single topic and are not easy to divide.
  3. We briefly mentioned thyroid-associated ophthalmopathy (TAO), idiopathic choroidal neovascularization, and arthritis-associated uveitis as examples of other ocular disorders that could be affected by sex hormones, although there is no evidence for these relationships to date.
  4. We have embraced the suggestion and added a figure presenting the ocular disorders influenced by sex hormones (Figure 1).

Reviewer 2 Report

It is a well-written review on sex hormones and their effects on ocular disorders. I found a few minor spelling mistakes (Page 5). I would suggest to add the important work of dr Ula Jurkunas on effect of estrogens on corneal endothelial cell survival (e.g. in Fuchs corneal endothelial degeneration) to the corneal section.

Author Response

An editing of English language and style was performed to correct spelling errors.